# Qualitative study investigating the perceptions of parents of children who failed vision screening at the age of 4–5 years

Alison Bruce,[1] Tom Sanders,[2] Trevor A Sheldon[3]

[1]Bradford Institute for Health Research, Bradford Teaching Hospitals NHS Trust, Bradford, UK
[2]School of Health and Related Research (ScHARR), Section of Public Health, University of Sheffield, Sheffield, UK
[3]Department of Health Sciences, University of York, York, UK

**Correspondence to**
Dr Alison Bruce; alison.bruce@bthft.nhs.uk

## ABSTRACT

**Objective** To explore in depth parents' experiences and understanding of their children's eye care in order to better comprehend why there is relatively low uptake of services and variable adherence to treatment.

**Design** Semistructured interviews, informed by the Health Belief framework, were conducted with parents of children who had failed vision screening at age 4–5 years. Four were parents of children who never attended follow-up, 11 had children who attended but did not adhere to spectacle wear and 5 parents of children who had attended and adhered. Interviews were recorded and transcribed verbatim; thematic analysis based on the constant comparative method was undertaken.

**Results** Parents' beliefs led to uncertainty about the benefit of treatment, with parents testing their children to confirm the presence of a vision deficit and seeking advice from other family and community members. The stigma of spectacle wear explained the resistance of some to their child's treatment with the maintenance of 'normality' often more important than clinical advice. The combination of parents' own health beliefs, stigma and the practicalities of attending appointments together influenced parental decisions. Attendance following vision screening and the decision to adhere to spectacle wear were primarily based on the perceived severity of the visual reduction with the perceived benefit of spectacle wear outweighing any negative consequences.

**Conclusions** Healthcare professionals require a greater understanding of parents' decision-making processes in order to provide personalised information. Knowledge of the cues to attendance and adherence provides policy makers a framework with which to review the barriers, develop strategies and redesign children's eye care pathways.

## INTRODUCTION

Decreased vision in young children is commonly due to the presence of refractive error, strabismus and/or amblyopia.[1] The presence of one or more of these conditions can result in a visual deficit in the developing infant. Most young children with refractive error or amblyopia do not however demonstrate obvious signs or symptoms, making it difficult for parents to identify, and vision screening is recommended

### What is already known on this topic?

- ► Reduced vision in young children is commonly related to refractive error, strabismus or amblyopia.
- ► Following identification of reduced vision at vision screening attendance for eye care appointments is poor with 30% of children failing to attend.
- ► Treatment consists of spectacle wear and may be combined with wearing an eye patch but adherence to treatment is inconsistent.

### What this study hopes to add?

- ► Reasons for non-attendance and non-adherence are complex with both pragmatic factors and health beliefs interlinked in parental decision making.
- ► The decision to adhere to spectacle wear was based primarily on the perceived severity of the visual reduction with the perceived benefit outweighing any negative consequences.
- ► The role of schools as facilitators should be considered when developing interventions to promote spectacle wear in young children.

in the United Kingdom at age 4–5 years.[2] Early detection programmes can only be effective if those identified with poor vision are appropriately treated. In deprived communities both locally[3] and internationally[4] attendance rates following vision screening are reported between 30% to 40%.[3] In addition, studies have shown poor adherence to spectacle wear[5–7]; therefore, a high proportion of children with identified needs do not access ophthalmic services and subsequent treatment.

Socioeconomic factors have been reported to influence access to children's eye care[8] and adherence with spectacle wear.[9] In USA, difficulty navigating the healthcare system, cost and loss and breakage of spectacles has been found to prevent attendance and adherence leading to inequalities in visual health.[10] Factors such as cultural sensitivity and health

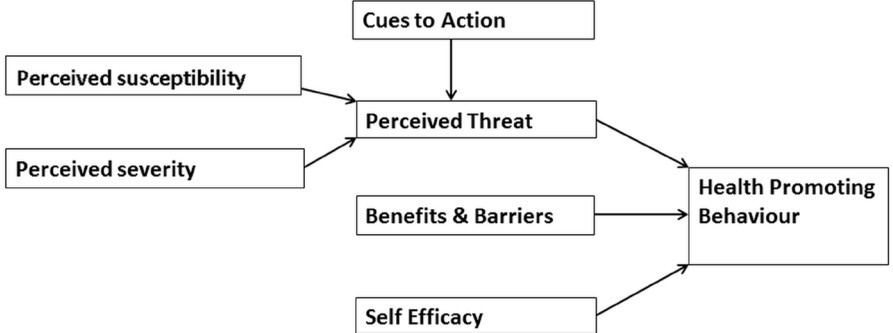

**Figure 1** Illustration of Health Belief Model.

beliefs have been associated with differences in access, uptake and adherence decisions in adults.[11] People's perception of the risk of health problems also influences healthcare utilisation.[12] Current literature therefore provides evidence of a number of factors impacting on access and uptake of children's eye services. However, unlike adult ophthalmic services[13] where patients' experiences have been documented, there is little reported qualitative evidence that would provide an understanding of parents' experiences of their child's eye care. This study explores in-depth parents' experiences and understanding of their children's eye care in order to better comprehend why there is relatively low uptake of services and variable adherence to treatment.

## METHODS
### Participant recruitment
The population-based vision screening programme in Bradford, a large multiethnic city in England, is offered to children in their year of school entry, aged 4–5 years. Children failing to achieve the pass criterion set by the UK National Screening Committee[2] (≤0.20 logarithm of the minimum angle of resolution (logMAR) in both eyes) are referred for examination, either to a community optometrist or the hospital eye service (HES) and subsequently followed up in order to measure the visual acuity (VA) with prescribed spectacles. As part of a separate longitudinal research programme,[14] children were followed up between 2013 and 2015 to explore the impact of adherence to spectacle wear on VA and early literacy. Parents of children participating in the study were invited for interview between November 2015 and May 2016. Parents were purposively selected if their child had failed vision screening in school years 2013–2014 or 2014–2015 and their child was either a non-attender (never attended follow-up appointments at their local optometrist or the HES) or their child was non-adherent (not wearing prescribed spectacles at unannounced visits in school and/or spectacle wear was not corroborated by documentation in the medical notes). For comparison purposes, parents of four children identified as adherent (wearing prescribed spectacles at unannounced visits in school) were also invited to participate.

Initially, 40 letters with prepaid reply slips were sent inviting parents (mother or father) of children participating in the longitudinal study to participate. No responses were obtained with this method. The letters were then distributed to the parents via their child's school. All interviews were undertaken in the school that their child was attending. Bradford is ranked the fifth highest deprived city in the UK; the five schools located across Bradford all had a Townsend score between 1 and 4 indicating lower socioeconomic status of the school communities. Four of the parents required interpreters (3 Urdu and 1 Parsi); the interpreters were teaching assistants working in the schools.

The study was informed by the Health Belief Model (HBM), a theoretical concept developed in the arena of public health (figure 1).[15] It consists of six key constructs that together provide an explanatory framework for the adoption of preventative health behaviour.[16] HBM was developed from the study of the factors influencing adults attendance at health screening programmes,[15] so it was chosen as a starting point to inform the topic guide for this study (table 1). The topic guide was used to ensure details of the parents' experiences and understanding of their child's eye care was captured, in particular their views on vision and spectacle wear in children and the influence of their own and the experiences of family members (table 1).

Semistructured interviews were conducted by one member of the study team (AB), a female orthoptist and postdoctoral research fellow, who was unknown to the parents, in the child's school at a time suitable for the parent, two mothers in one school arrived at the same time and a combined interview was undertaken. Both the mother and father of one child were interviewed together; the father was the main contributor; therefore, only his discourse is included in the analysis. Interviews were audio recorded and transcribed verbatim; in addition, the non-English transcripts were listened to by a bilingual research assistant to confirm accuracy of the interpretation.

Two members of the study team (AB and TS) reviewed the transcripts independently and then together developed a thematic analysis, using constant comparison between the transcripts, identifying key themes and concepts.[17 18] Our approach to analysis was based on

**Table 1** Topic guide (based on the Health Belief Model[14])

| Concept | Definition | Areas of questioning |
|---|---|---|
| Perceived susceptibility | An individual's assessment of their chances of developing a condition. | Experience of eye conditions encountered in children. Experience of eye tests and or treatment as a child/adult. Eye care advice received from parents/friends. Advice they would give their child. |
| Perceived severity | An individual's opinion as to the seriousness of the condition. | Importance of vision: worries, fears and attitudes. Attitudes to children wearing glasses. Attitudes to children wearing eye patches. Knowledge of impact of poor vision on employment, driving and other everyday activities. |
| Perceived benefits | An individual's opinion as to whether a new behaviour is better than current behaviour. | Importance of good vision. Impact of poor vision in children. Impact of vision on a child's ability to learn. |
| Perceived barriers | An individual's opinion as to what will prevent them from adopting a new behaviour. | Costs and relative costs. Satisfaction with health services. Number of visits to optometrist or hospital. Obstacles to attending. Eye clinic opportunities. Cosmetic impact. |
| Self-efficacy | Belief in one's own ability to perform an action. | Capability of arranging appointments. Knowledge of children's eye tests. Capability of performing the treatment (ensuring your child wears glasses or eye patch every day to improve their vision). |
| Cues to action | Factors that will prompt a person into changing behaviour. | Family attitude to eye care. Attitude of school to eye care. Any associated cultural or religious practices in relation to children's or adult eye care. |

the 'constant comparative' methodology derived from grounded theory. We analysed the interviews in search of similarities and differences, which led to the development of themes in the data. Thematic analysis was chosen to allow for the emergence of opinions that were not anticipated in advance. No member checking was performed with this hard-to-recruit group of parents; however, a focus group with a small number of mothers confirmed the interpretation of the findings. Written consent was obtained prior to each interview.

## RESULTS

In-depth interviews were completed with 20 parents (17 mothers and 3 fathers) of 19 children from five schools; 14 parents were South Asian, 5 were white British and 1 African heritage (table 2). No further interviews were undertaken after saturation of the emerging themes from the 19 interviews. Our analysis identified themes that highlighted pragmatic reasons for non-attendance and non-adherence and also those related to health beliefs about illness, disease and consultation behaviour; illustrative quotes are presented in the text. The themes identified present the complexity of parental decision making.

## Health beliefs

Deep-rooted health beliefs played an important part in explaining attitudes towards vision care in children

starting primary school. Doubt along with other cultural health beliefs gave rise to uncertainty surrounding the potential benefits of spectacle wear, viewed by some as unnecessary.

### Confidence in the test

Reduced vision in young children is not generally associated with overt signs and symptoms, and parents found it difficult to accept their child had a vision problem. Adherence to spectacle wear was supported by parents when they perceived the vision test to be reliable.

It was really hard when she was young and it was the first time she had her eyes tested and they were putting some things in front of her eyes and I think she found that difficult. (Mother 3, adherent)

You do find the children do get bored very, very quickly but they dealt with them in a pleasant manner but sometimes it makes me wonder whether they actually could tell what she was seeing properly because, with them being so young. (Mother 6, adherent)

He's never had any problems with his work to say that he's got bad eyes. (Mother 10, non-attender)

**Table 2**  Participant characteristics

| Parent | Ethnicity | Attended and adherent | Attended and non-adherent | Non-attendance and non-adherent |
|---|---|---|---|---|
| Mother 1 (i) | South Asian | | | Yes |
| Mother 2 (i) | South Asian | | Yes | |
| Mother 3 | South Asian | Yes | | |
| Mother 4 | African | | Yes | |
| Mother 5 | South Asian | | Yes | |
| Father 1 | White British | Yes | | |
| Mother 6 | White British | Yes | | |
| Mother 7 | South Asian | | Yes | |
| Father 2 | South Asian | | Yes | |
| Mother 8 | South Asian | | Yes | |
| Mother 9 | White British | | Yes | |
| Mother 10 | White British | | | Yes |
| Mother 11 | White British | | | Yes |
| Mother 12 | South Asian | | Yes | |
| Mother 13* | South Asian | | | |
| Father 3 | South Asian | Yes | | |
| Mother 14 (i) | South Asian | | Yes | |
| Mother 15 | South Asian | | Yes | |
| Mother 16 | South Asian | | Yes | |
| Mother 17 | South Asian | | | Yes |

*Mother 13 and father 3 are parents of the same child and were interviewed together; only the fathers discourse is reported.
i, interpreter.

As he's got a bit older and he's obviously done more eye tests ….I'm a lot happier that (you know) he is properly diagnosed. (Mother 8, non-adherent)

The view that children 'should not' require spectacles and the lack of confidence in the accuracy of testing young children led to parents testing their own children at home and canvasing significant others to verify that the child's vision was good without spectacles.

Mums tried her at home to check & test to see if she needs them but there is no difference so it's not like she needs them at home at all. (Mother 3, adherent)

She does gymnastics as well and she doesn't wear her glasses and she can see on the beams and the floor and everything fine. (Mother 6, adherent)

…. she's fine at home she doesn't need the glasses at home. She goes to the mosque as well and she will be doing reading there as well and she doesn't need them at mosque. She's asked the teacher there as well and she says she's fine and she doesn't need them there as well. (Mother 2, non-adherent)

I says to her can you see without them and she says yeh I can see without them, yeh… she's had no prob-lems at all, she can see alright, fine. She's had no pain in her eyes or anything like that, she says she doesn't really need them she can see everything without them. (Father 2, non-adherent)

## Community and social influences
The advice of family or community members was influential, particularly when it matched parents' personal beliefs. Where there was a positive family history of wearing spectacles parents reported attending and adherence with spectacle wear.

Well you do have more pressure from older people like my parents, 'why is he wearing glasses, his eye sight is going to get worse, he'll get dependant on them'. (Mother 8, non-adherent)

My mum has said it, like they're too young… I do have this from all my family, …. he is obviously too young and once he wears them they're going to be like you're going to have to wear them all the time. (Mother 17, non-attender)

He (father) doesn't feel that there is a problem so that's the reason why he hasn't taken her (child). (Mother 1, non-attender)

No one has discouraged the children from wearing glasses, everybody has asked … does she need glasses is that why she's wearing them…. so she gets a lot of encouragement. (Mother 3, adherent)

She was quite happy actually, because she's wearing them (glasses) like her older sister and brother and daddy…. (Mother 12, non-adherent)

### Understanding

Parents of those children who were adhering to spectacle wear believed that failure to wear the spectacles would lead to a deterioration of both the child's vision and their ability to learn, and many parents believed that by wearing the spectacles, the vision would improve and then spectacles would no longer be required.

…. if he takes them off then his vision is going to get really weak. Glasses will help him so he has to keep them on all the time and just take them off when he really has to. (Mother 3, adherent)

… if a kid cannot see properly and keeps getting headaches because you know he's trying to concentrate, it's going to affect him in his education, it's going to affect him when he's playing and during life as well, so it's very important I think. (Father 3, adherent)

… if she keeps wearing them now that when she gets to a teenager she might not need them. (Mother 6, adherent)

### Stigma

The stigma of spectacle wear was reported by parents of adherent and non-adherent children, and this contributed significantly in explaining parents' resistance to the child's spectacle wear.

To be honest yesterday she took her glasses out of school and she thought oh I'm a very different girl, my friends are saying you're a different […] to be honest she doesn't need really, she's OK without glasses. (Mother 5, non-adherent)

… in Zimbabwe people who wear glasses they are seen as people who are really well educated. They say, 'You're wearing glasses, you think you're sooo… educated'. (Mother 4, non-adherent)

… because they're teasing him he's retaliated and we've had a few issues. (Father 3)

### Pragmatism

Access to appointments was influenced by the parent's ability to organise appointment times around busy working schedules, the children's school attendance and the availability of transport.

Awkward, really awkward because it's generally in school time and she doesn't like missing any time at school…. (Mother 6, adherent)

When I was at work it was difficult obviously because you know when you're at work you can't really get the appointments during the day and all the kids are at school…(Mother 8, non-adherent)

No, it's just main problem is parking, I just missed two appointments because I couldn't get the parking on time. (Mother 7, non-adherent)

### Information

The amount and detail of information provided to parents varied and parents suggested that more information was always helpful. Two mothers suggested that the eye care professional should provide parents with an indication of the improvement in VA at each visit.

We've always had leaflets given to us every time we've been so we were given plenty of information. (Mother 6, adherent)

I wasn't fully clear of the letter and what information I did take from it. (Mother 1, non-attender)

They (eye professionals) need to understand a little better and give a percentage, so we (parents) can compare vision to the last test. (Mother 16, non-adherent)

It would be good to be updated on whether her eyesight was getting better or worse. (Mother 2, non-adherent)

Parents whose first language was not English did not always fully comprehend the information provided or relied on family members to access care and interpret clinical advice.

… my older daughters can read now so that they can read and interpret letters. (Mother 3, adherent)

I used to take my sister in law, who can understand English, so she can explain it and ask questions. (Mother 14, non-adherent)

### Supporting strategies

The children's schools played a key role in supporting the parents in developing strategies to both attend ophthalmic appointments and ensure the childs spectacle

wear. Some parents were able to negotiate appropriate time for their child to be absent during school hours.

> They were fine, the times that they gave me have always been in the afternoon instead of the morning so it's easy… and I can take them out of school. (<other 2, non-adherent)

> So (erm), many times many occasions I've forgot glasses at home and then I gave one pair to school as well to resolve this issue. (Mother 12, non-adherent)

> My son needs [glasses] all the time, so I have to say all the time to class teacher 'please remind him to wear his glasses'. (Mother 16, non-adherent)

## DISCUSSION

The decision to attend following vision screening and to adhere to spectacle wear was primarily based on the perceived severity of the visual reduction with the benefit of spectacle wear outweighing any negative consequences. Parental observations played an important role in validating the professional assessment, when parental perception did not match clinical opinion adherence was less likely.[19] Parental knowledge tended to dominate decisions where there was discordance between professional and lay knowledge[20] with parents having difficulty accepting clinical advice, particularly when their child did not appear to require spectacles for reading and writing. It has been reported that limited experience of a health condition affects judgement of its severity[21]; this may also have influenced the parents' reasoning to support spectacle wear in their children. Parents whose children had failed vision screening and had not taken their child for further assessment were keen to justify their actions by normalising the eye condition[19 22]; in addition, they did not report any support from either family members or from their child's school; this lack of support may have contributed to non-attendance. Stigma played a major part in explaining parents' resistance to clinical advice.[23] Appearing 'normal' seemed to outweigh potential benefits[24] and was a strong driver for maintaining a socially acceptable image of the family within the community[22]; a child with spectacles seemed to threaten this image suggesting that the eye condition was not the sole factor impacting on adherence.[25] Parents who did not consider their child's vision to be impaired, who described barriers such as organising transport or social stigma, did not follow recommendations and deemed treatment unnecessary. Parents with limited English language ability relied on family members to access care and interpret clinical advice; this use of the family network is reported to be unreliable, leading to misinformation and has been reported to impact on the child's care.[12] The practical considerations of attending regular appointments particularly in cases of uncertainty about the benefit of treatment therefore posed a dilemma, contributing to and perhaps being the tipping point for non-attendance and non-adherence. Where support was available either from family members or from the child's school, this provided a cue to action prompting parents to take the first step in attending appointments.

### Strengths and limitations

This qualitative study helps to explain the reasons for non-attendance and poor adherence to spectacle wear in young children following vision screening. The strengths of the study include the purposive sample from a multi-ethnic community, including parents whose children were known to fail to attend and those whose children failed to adhere to prescribed spectacle wear; these are hard-to-reach groups of parents. The additional inclusion of parents whose children had attended and adhered to spectacle wear allowed insights into both barriers and enablers to attendance and adherence.

The children were participants in the Born in Bradford (BiB) cohort, and this could have encouraged positive responses in recruitment and in the views provided; however, no parent responded solely from the initial invitation to participate generated via BiB. The schools were crucial to the successful recruitment of parents, and this may have positively influenced the parental responses regarding the support provided by schools; however, all the parents declined to be interviewed in their own home and would only be interviewed in school. The participating parents were all recruited from deprived communities, and this study may not be representative of the experiences of parents in more affluent areas; however non-attendance and non-adherence is known to occur predominantly in areas of lower socioeconomic status,[26] and therefore these findings will represent the experiences of this under reported, hard-to-reach group of parents.

### Existing literature

The findings from this study provide a comprehensive picture of the experiences of parents whose young children have been referred following vision screening, further explaining the reasons for non-attendance and non-adherence. Previous population-based studies reporting children's adherence to spectacle wear have focused on the factors found to be associated with non-attendance, reporting higher proportions of children from families of lower socioeconomic status[26] failing to attend and older children demonstrating a greater rate of non-adherence to spectacle wear.[27] In a study of young children (under 8 years) in Berkshire,[28] compliance with spectacle wear was reported to be good; however, the place where the study was undertaken is in a relatively affluent area, and the parents were regular attenders to the HES.

Similar to our findings, studies of the experiences of adults in screening programmes describe that a lack of perceived severity of the condition influences

attendance.[13 29] This study, in the field of children's eye care, adds to this, reporting that where the severity of the condition is in doubt, the parent will perform their own confirmatory vision test in their home environment. The practical barriers to attendance we report such as access to appointments and transport also confirm the findings of previous studies.[13 29] In addition, importantly, this study highlights the influence that support from significant others such as family or school have in helping overcome barriers to attendance and adherence.

## Implications for clinicians and policy makers

Pragmatic strategies are required to improve attendance and adherence. The desire for condition-specific information is underestimated by clinicians,[30] and provision of personalised information, following vision screening, highlighting the benefits of treatment in young children, could improve attendance. The current national development of information leaflets specifically aimed at parents providing information following vision screening may influence attendance.[31] Concordance around treatment should involve discussion between clinician and parent through which an informed decision regarding treatment can take place.[32] This has been reported in general practice with patients less likely to take medications if their own concerns are not initially addressed.[20] Further study is required into how eye care professionals communicate with parents, and how they present information[33] to ensure the parents can make informed decisions regarding adherence. Stigma was commonly cited by parents as a barrier to adherence.[34] Strategies both at the individual and community level are required with the role of schools as facilitators in reducing stigma and promoting adherence, an area highlighted by parents and requiring future study.

These findings provide insight into the reasons parents either rejected or resisted therapeutic advice, not solely reflecting levels of knowledge, but reflecting an active evaluation of the potential severity of the vision loss and the perceived loss of normality and social stigma compared with the benefit of the treatment. Our results illustrate the complexity around attendance and adherence patterns and provide a greater understanding that can inform the redesign of children's eye care pathways.

**Acknowledgements** We would like to thank all the families and schools who took part in this study.

**Contributors** AB initiated the project, designed the study, conducted the interviews, analysed the transcripts and drafted and revised the paper. She is guarantor. TS analysed the transcripts and revised the draft paper. TAS initiated the project and revised the draft paper.

**Funding** AB is funded by a National Institute for Health Research Post-Doctoral Fellowship Award (PDF-2013-06-050). The research was funded by the NIHR Collaboration for Leadership in Applied Health Research and Care Yorkshire and Humber (NIHR CLAHRC YH). .The views and opinions expressed are those of the author(s), and not necessarily those of the NHS, the NIHR or the Department of Health.

**Disclaimer** The views expressed are those of the author(s) and not necessarily those of the NHS, the NIHR or the Department of Health.

**Competing interests** None declared.

**Patient consent** Not required.

**Ethics approval** Ethics approval was obtained from the National Research Ethics Committee Yorkshire and the Humber- South Yorkshire UK (Ref 13/YH/0379).

**Provenance and peer review** Not commissioned; externally peer reviewed.

**Data sharing statement** No additional unpublished data are available.

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
