## [Reviewer comments · BMJ Paediatrics Open]

This paper was submitted to a another journal from BMJ but declined for publication following peer review. The authors addressed the reviewers' comments and submitted the revised paper to BMJ Paediatrics Open. The paper was subsequently accepted for publication at BMJ Paediatrics Open.

ARTICLE DETAILS

TITLE (PROVISIONAL)	A qualitative study investigating the perceptions of parents of children who failed vision screening at the age of 4-5 years.
AUTHORS	Bruce, Alison; Sanders, Tom; Sheldon, Trevor

VERSION 1 - REVIEW

REVIEWER	Reviewer Name: Liu, Xiaoxuan Reviewer Affiliation: Queen Elizabeth Hospital Birmingham No competing interests
REVIEW RETURNED	04-Feb-2018

GENERAL COMMENTS	The paper addresses an important issue of non-adherence and non-attendance for prescribed childhood spectacle wear. The methodology used is appropriate and the findings reveal a range of factors which may influence parents' health beliefs and decisions in this context. These findings are applicable to clinical practice in a real-world setting.
---

VERSION 1 – AUTHOR RESPONSE

We thank you for your comments and recognition that the findings reflect the decision making process of parents whose young children have been prescribed spectacles following vision screening.

VERSION 2 – REVIEW

REVIEWER	Reviewer name: Bronia Arnott Institution and Country: Newcastle University, UK Competing interests: None
REVIEW RETURNED	07-Jun-2018

GENERAL COMMENTS	This manuscript aimed to explore parental reasons for non-attendance at follow up appointments and non-adherence to spectacle wear following a childhood vision screening which had identified a vision problem. Parents of 19 children with vision problems participated in semi-structured interviews informed by the Health Belief Model. Families were purposively sampled to include non-attendees, non-adherers, and those who attended and adhered. These families were recruited from an area of socio-economic deprivation and were ethnically diverse and therefore this research speaks to populations which are often absent from academic research.
---

Results showed that the main issues related to: parental perceptions of the severity of the problem (and questioning if there was a problem at all); and parental concerns about stigma relating to wearing spectacles.

There are a number of issues for attention which are detailed below.

Overall comments

1. Firstly the authors are directed to the COREQ <http://www.equator-network.org/reporting-guidelines/coreq/> a 32 item checklist for reporting qualitative research. The authors should ensure that they are reporting on all of the necessary elements. A particular area of concern was the lack of current clarity regarding relationship with the participants, data collection, and data analysis. Recommend that the authors go through each of the items and ensure that they report the information as required.

2. A figure should be included to illustrate the Health Belief Model.

Specific Comments

Title

1. I was concerned that the title was not a good representation of the paper. Whether children were too young for spectacles was only one of the issues that raised by parents – and indeed the two main issues seemed to be related to severity and stigma. The title could be changed to more appropriately reflect the study.

2. Further I wasn't convinced that 'qualitative review' was the best descriptor for the study design – this would indicate that it was a review of existing qualitative studies. This should be revised to describe the study as a 'qualitative study'.

Background

1. I would expect the background to discuss whether there was any qualitative research in this area already or if there was any qualitative research in a related area. The background section should be updated.

2. I would also expect to see the theoretical approach to the study introduced in the background section. The study is informed by the Health Belief Model and therefore there should be some introduction to this model, including justification for selection and critical review in comparison to alternatives. There is also no mention of the specific epistemological approach taken and this should be added.

Method

The method section would benefit from more detail and justification. It could also be re-organised so that the information is presented in a more coherent, chronological manner.

1. The definition of adherence is quite narrow – observation of adherence during follow-up appointments. Could it be that some children did wear their spectacles at follow-up appointments but did so at other times? Could it be that the type of parents who would encourage their children to wear spectacles to follow up appointments (but were usually non-adherent) be the parents who would stress the benefits of spectacle wearing in interviews?

However, the measure of adherence is objective and therefore the study benefits from not having to rely solely on parent/child report – but it could have been supplemented by self-reports. The operationalisation of adherence should be justified and the advantages and disadvantages discussed.

2. Related to this, the definition of non-attendance is unclear. How many appointments had these families failed to attend to be considered non-attendees? Please include this information.

3. Recruitment of families could be clearer. It is not clear how many were approached to participate and what proportion consented. It is not clear if the approach was made face to face, by letter, or over the phone. More information should be added.

4. The sample size is also an issue. Specifically, the sub-groups (which are quite heterogeneous) are small and there is no discussion as to why recruitment did not continue to recruit more families in each subgroup. Justification should be provided for this decision.

5. There is no mention of data saturation. Was this something which was assessed? Was there an ongoing assessment which influenced when recruitment was stopped? Also did the data include 'thick description'? These issues should be discussed.

6. Was the topic guide iterative or fixed? Did the line of questioning develop as the interviews were conducted or did it remain the same throughout? Please add this information to the manuscript.

7. How long after the eye test did these interviews take place? There is some mention that the families were followed for 2 years but it wasn't clear at what point during the 2 years these interviews took place. Was it the same length of time following the test for everyone? Please add this information.

8. Given that the Health Belief Model informed the development of the topic guide for the interviews I wondered why the authors did not use framework analysis based on the HBM but chose thematic analysis. Could the authors add their justification for this.

9. There was no member checking – either of interpretation of the interviews or in the analysis of the results. The authors should discuss why they chose not to include member checking.

Results

1. I am somewhat surprised that even although the authors did not employ framework analysis (a deductive approach), the themes that emerged from the interviews are shown to 'fit' into the HBM framework through inductive analysis. The authors should discuss this.

2. I have concerns about the thematic analysis and the way that some quotations have been analysed. For example there is a comment which has been coded as 'perceived susceptibility' which refers to the child being scared of the eye test. There is another which has been coded as self-efficacy which relates to the parents not receiving the results from the eye test. It may be that with further context the coding of the quotes would become clearer. If not there would need to be some justification as to how they meet the definition of the theme.

3. I would expect to see at least 3 quotes presented per theme. If there are not sufficient quotes the authors should consider whether the theme has a higher order theme that it would fit with.

4. There were quotes relating to eye patches in the results section and yet the use of eye patches had never been mentioned at any point previously in the manuscript.

	This would be a useful addition to orient the reader. 5. There could be reflections on the qualitative process and whether there were any factors which could have influenced the results. For example, the interviewer was an orthoptist – were parents aware of this? Did they think that this person could influence their cared depending on what they said during the interview? Could this have influenced the results? Discussion 1. There are a number of issues highlighted in this review which may need to be included as limitations in the discussion section. 2. The authors talk about the limited English of some participants in the discussion and how they relied on family to access care and interpret clinical advice – yet this does not appear in the results section but seems important.
--	---

REVIEWER	Reviewer name: Jennifer McAnuff Institution and Country: NIHR Clinical Doctoral Research Fellow, Institute of Health and Society, Newcastle University, UK Competing interests: None to declare
REVIEW RETURNED	17-Jun-2018

GENERAL COMMENTS	Thank you for the opportunity to review this interesting manuscript. I don't have expertise in this area of healthcare practice; therefore, my review focuses on the qualitative methods used, and how these have been reported. I'd like to raise a few questions about aspects of the manuscript and make some recommendations for revisions. One these have been addressed, I think the paper will be a relevant and useful publication in BMJ Paediatrics Open. Title: I can see that the title stems from one aspect of the results (e.g. lines 17-25, page 7/19). However, I don't see why this particular aspect has been used in the title, when it does not stand out/has not been proposed by the authors as the most important result. For example, it doesn't feature in the abstract. I recommend the title is revised accordingly. Abstract: If possible, it would be useful to specify the theoretical framework in the methods section. My questions and recommendations below, if accepted, will lead to further changes in the abstract. Introduction: Line 17, page 3/17. The statement that 30% of children fail to attend follow-up is based on one study in Bradford, which may or may not be generalisable. Therefore, I recommend that the authors provide some brief context to clarify (the limits of) what is known about the proportion of children who do not attend follow-up. Could the study aim be consistently stated between the abstract and the introduction?
---

Methods:

Some clarification regarding the sampling, recruitment strategy, and response rates would be helpful. What was the sampling frame (all the schools, or a sub-section?), how many parents were invited to take part (if known), how were they approached, what proportion responded, why was 20 an appropriate number etc.

The remainder of the methods section is very light on detail about data analysis. How was the theoretical framework applied in data analysis, or did it only guide data collection? What were the steps in the thematic analysis? The abstract mentions constant comparison – how was this integrated with the thematic analysis?

Results:

The authors seem to be presenting the results in three ways: (1) By sub-group (those who don't attend/don't adhere, attend/don't adhere, attend/adhere), (2) according to the six constructs in the theoretical framework (i.e. Tables 2-3), and (3) in terms of themes arising from the data, and broadly related to the theoretical framework (e.g. in the abstract, and on pages 9 and 10). I found this confusing, and at times superficial (particularly 1-2).

I was expecting the results to be presented in terms of the constructs within the theoretical framework.

Could the authors clarify how the reader should 'use' the results section and the two tables? E.g. how the cross-referencing between the three should be done?

Table 3 repeats the quotes in the results section. I can see that it also presents them in terms of the theoretical constructs, so perhaps that is what Table 3 adds. However, this is fairly superficial and could be more convincing. For example – I didn't see how the quotes presented as illustrating self-efficacy actually related to that theoretical construct.

The results section reads like a list, which I think contributes to it appearing superficial. Perhaps it would be helpful to integrate the theoretical contents of table 3 into the text?

Line 29, page 7 – I recommend using alternative wording to 'claimed', as it implies an inappropriate/unwarranted scepticism about what those parents said.

Line 27, page 8: "...the benefits of spectacle wear were only reported by parents whose children were adherent." This seems self-evident, I wasn't sure what point the authors were trying to make.

Discussion

The first two pages of the discussion section are actually results (with a few discursive references to wider literature). I think they are the more interpretive, cross-cutting themes that I was wanting above. I recommend that they should be integrated with the results section, bearing in mind the points I have made above. (In other words, the content may need to be re-worked in light of my feedback.)

I don't understand why shared decision making suddenly appears on line 33, page 9 – are the authors over-reaching on what they can claim from their results here?

	Lines 44-49, page 9. The sentence about parents not appearing to take responsibility is: (a) over-reaching what can be claimed from the data (unless you're reporting they explicitly said it's not their responsibility?), and (b) unfortunately worded, in that it sounds like you are blaming parents (which I'm sure is not the case). Could this be re-worded to clarify the analytical/interpretive point being made? I'm struggling with several uses of 'more likely, less likely', and the general idea of reporting these results in terms of 'likelihoods' (i.e. quantitative connotations, over-stretching what can be interpreted from the data...). Maybe this is inappropriate given the methodology, or maybe it will materialise as entirely appropriate once additional detail is provided on the methods in the study. I would like to see a more thorough of strengths and weaknesses at the beginning of the discussion section. Then, more broadly, I'd like to see the discussion section flow more directly from the results, i.e. health beliefs, stigma, and practicalities. For example, the aspects on information provision, communication etc should be discussed in the context of those three main 'themes' in the results. The structure of the discussion could be clearer, could the authors use the BMJ Paediatrics Open authors guidance? "Discussion: we recommend, but do not insist, that the discussion section is no longer than five paragraphs and follows this overall structure (you do not need to use these as subheadings): a statement of the principal findings; strengths and weaknesses of the study; strengths and weaknesses in relation to other studies, discussing important differences in results; the meaning of the study; possible explanations and implications for clinicians and policymakers; and unanswered questions and future research." Thank you again, and I look forward to reviewing a re-submission, if required.
--	---

VERSION 2 – AUTHOR RESPONSE

Reviewer: 1

This manuscript aimed to explore parental reasons for non-attendance at follow up appointments and non-adherence to spectacle wear following a childhood vision screening which had identified a vision problem. Parents of 19 children with vision problems participated in semi-structured interviews informed by the Health Belief Model. Families were purposively sampled to include non-attendees, non-adherers, and those who attended and adhered. These families were recruited from an area of socio-economic deprivation and were ethnically diverse and therefore this research speaks to populations which are often absent from academic research. Results showed that the main issues related to: parental perceptions of the severity of the problem (and questioning if there was a problem at all); and parental concerns about stigma relating to wearing spectacles.

There are a number of issues for attention which are detailed below.

Overall comments

1. Firstly the authors are directed to the COREQ <http://www.equator-network.org/reporting-guidelines/coreq/> a 32 item checklist for reporting qualitative research.

The authors should ensure that they are reporting on all of the necessary elements. A particular area of concern was the lack of current clarity regarding relationship with the participants, data collection, and data analysis. Recommend that the authors go through each of the items and ensure that they report the information as required.

Thank you for highlighting the COREQ guidelines, in the revised paper we have utilised the guidance to ensure that all required elements are reported. In particular we have provided greater detail regarding the participants (pages 4, 5 & 6, Table 2), data collection (pages 4 & 5) and data analysis (page 5). A revised COREQ checklist has been submitted.

2. A figure should be included to illustrate the Health Belief Model.

A figure to illustrate the Health Belief Model is now included (Figure 1).

Specific Comments

Title

1. I was concerned that the title was not a good representation of the paper. Whether children were too young for spectacles was only one of the issues that raised by parents – and indeed the two main issues seemed to be related to severity and stigma. The title could be changed to more appropriately reflect the study.

We have included a more generic title to reflect the range of findings:

“A qualitative study investigating the perceptions of parents of children who failed vision screening at the age of 4-5 years.”

2. Further I wasn't convinced that 'qualitative review' was the best descriptor for the study design – this would indicate that it was a review of existing qualitative studies. This should be revised to describe the study as a 'qualitative study'.

The online submission category will be changed on resubmission of the revised paper.

Background

1. I would expect the background to discuss whether there was any qualitative research in this area already or if there was any qualitative research in a related area. The background section should be updated.

The introduction has been updated accordingly and now includes review of the qualitative research in the area (page 3, line 65 – 79).

2. I would also expect to see the theoretical approach to the study introduced in the background section. The study is informed by the Health Belief Model and therefore there should be some introduction to this model, including justification for selection and critical review in comparison to alternatives. There is also no mention of the specific epistemological approach taken and this should be added.

We have added a short statement in the methods section indicating the epistemological approach adopted in the study. Although the Health Belief Model provides the theoretical framework informing the interpretation, we also adopted 'constant comparative' methodology informed by grounded theory to develop themes from the data.

Method

The method section would benefit from more detail and justification. It could also be re-organised so that the information is presented in a more coherent, chronological manner.

1. The definition of adherence is quite narrow – observation of adherence during follow-up appointments. Could it be that some children did wear their spectacles at follow-up appointments but did so at other times? Could it be that the type of parents who would encourage their children to wear spectacles to follow up appointments (but were usually non-adherent) be the parents who would stress the benefits of spectacle wearing in interviews? However, the measure of adherence is objective and therefore the study benefits from not having to rely solely on parent/child report – but it could have been supplemented by self-reports. The operationalisation of adherence should be justified and the advantages and disadvantages discussed.

The methods section now includes greater detail of the recruitment, interviews and analysis process. A more detailed explanation of the definitions of non-attendance, non-adherence and adherence is also now included (page 4, 88 – 103).

2. Related to this, the definition of non-attendance is unclear. How many appointments had these families failed to attend to be considered non-attendees? Please include this information.

Families considered to be non-attenders had never attended any appointment following referral from the vision screening programme (page 4, 97 – 98).

3. Recruitment of families could be clearer. It is not clear how many were approached to participate and what proportion consented. It is not clear if the approach was made face to face, by letter, or over the phone. More information should be added.

A clearer account of the recruitment process is included with detailed explanation of the numbers recruited (page 4 line 104-111).

4. The sample size is also an issue. Specifically, the sub-groups (which are quite heterogeneous) are small and there is no discussion as to why recruitment did not continue to recruit more families in each subgroup. Justification should be provided for this decision.

Thematic analysis showed that the emerging themes were not unique to any one group of parents and saturation was reached after the 19 interviews (page 6, line 144-145). We have now stated this in the paper along with the difficulties in recruiting (page) this hard to access group of parents.

5. There is no mention of data saturation. Was this something which was assessed? Was there an ongoing assessment which influenced when recruitment was stopped? Also did the data include 'thick description'? These issues should be discussed.

Data saturation is now reported (page 6, 144 – 145).

6. Was the topic guide iterative or fixed? Did the line of questioning develop as the interviews were conducted or did it remain the same throughout? Please add this information to the manuscript.

The topic guide was used to inform the discussion and ensure that key areas of interest to the research study were included in the interview. The interview style, however, did encourage participants to elaborate in more detail the issues of greatest interest to them. This is now stated in the manuscript (page 5 line 135 - 136).

7. How long after the eye test did these interviews take place? There is some mention that the families were followed for 2 years but it wasn't clear at what point during the 2 years these interviews took place. Was it the same length of time following the test for everyone? Please add this information.

This information is now included in the manuscript (page 4, 93 – 96).

8. Given that the Health Belief Model informed the development of the topic guide for the interviews I wondered why the authors did not use framework analysis based on the HBM but chose thematic analysis. Could the authors add their justification for this.

The Health Belief Model was used as the starting point to inform the topic guide as this is a relatively under-reported area of research and we did not wish to miss relevant information. The HBM however has mainly been reported in the field of adult screening programmes and we did not want to restrict emerging themes arising from the parental interviews. Thematic analysis was therefore chosen since it allows for the emergence of opinions that were not anticipated in advance (page 5 line 113 – 121).

9. There was no member checking – either of interpretation of the interviews or in the analysis of the results. The authors should discuss why they chose not to include member checking.

Member checking was not included as this was a relatively hard-to-recruit group of parents and it was unlikely that parents would have consented to further interviews. AB did however, feedback to a small focus group of mothers who confirmed the interpretation of the interviews and further suggested greater inclusion of schools in supporting with children's spectacle wear (page 5 line 136 - 138).

Results

1. I am somewhat surprised that even although the authors did not employ framework analysis (a deductive approach), the themes that emerged from the interviews are shown to 'fit' into the HBM framework through inductive analysis. The authors should discuss this.

Both reviewers have commented on the incongruous presentation of the results and how they do not quite fit into the HBM constructs in Table 3. The results were analysed using thematic analysis, however, in order to present parental quotes and comply with the word limit, the quotes were presented in Table 3. We recognise that this is confusing for the reader and also as the results stemmed from the thematic analysis they do not present an exact fit for the HBM constructs. We have therefore removed Table 3 from the paper and present the quotes in the body of the text. We trust that the increased word count is acceptable to both the reviewers and the editors.

2. I have concerns about the thematic analysis and the way that some quotations have been analysed. For example there is a comment which has been coded as 'perceived susceptibility' which refers to the child being scared of the eye test. There is another which has been coded as self-efficacy which relates to the parents not receiving the results from the eye test. It may be that with further context the coding of the quotes would become clearer. If not there would need to be some justification as to how they meet the definition of the theme.

As per point 2 above the results have now been presented in terms of the three main themes and the sub-themes that arose from the thematic analysis.

3. I would expect to see at least 3 quotes presented per theme. If there are not sufficient quotes the authors should consider whether the theme has a higher order theme that it would fit with.

A minimum of three quotes per theme have now been included.

4. There were quotes relating to eye patches in the results section and yet the use of eye patches had never been mentioned at any point previously in the manuscript. This would be a useful addition to orient the reader.

The quote regarding eye patching has been removed from the text and replaced with a more appropriate quote.

5. There could be reflections on the qualitative process and whether there were any factors which could have influenced the results. For example, the interviewer was an orthoptist – were parents aware of this? Did they think that this person could influence their cared depending on what they said during the interview? Could this have influenced the results?

The principal researcher was an orthoptist by professional background but was not known by the participants. This has been added to the text (page 5, line 122 -123).

Discussion

1. There are a number of issues highlighted in this review which may need to be included as limitations in the discussion section.

The Strengths and Limitations section has been extended (page 13, line 342 – 360).

2. The authors talk about the limited English of some participants in the discussion and how they relied on family to access care and interpret clinical advice – yet this does not appear in the results section but seems important.

In the revised presentation quotes have now been provided in the results section “Information” (page 11 line 287 - 294).

Reviewer: 2

Comments to the Author

Thank you for the opportunity to review this interesting manuscript. I don't have expertise in this area of healthcare practice; therefore, my review focuses on the qualitative methods used, and how these have been reported.

I'd like to raise a few questions about aspects of the manuscript and make some recommendations for revisions. One these have been addressed, I think the paper will be a relevant and useful publication in BMJ Paediatrics Open.

Title:

I can see that the title stems from one aspect of the results (e.g. lines 17-25, page 7/19). However, I don't see why this particular aspect has been used in the title, when it does not stand out/has not been proposed by the authors as the most important result. For example, it doesn't feature in the abstract. I recommend the title is revised accordingly.

The title has been revised accordingly to that stated above.

Abstract:

If possible, it would be useful to specify the theoretical framework in the methods section.

The Health Belief Model has now been stated in the abstract (page 2, line 36).

My questions and recommendations below, if accepted, will lead to further changes in the abstract.

Introduction:

Line 17, page 3/17. The statement that 30% of children fail to attend follow-up is based on one study in Bradford, which may or may not be generalisable. Therefore, I recommend that the authors provide some brief context to clarify (the limits of) what is known about the proportion of children who do not attend follow-up.

Further information and references have been added to the introduction (page 3 line 65 – 67).

Could the study aim be consistently stated between the abstract and the introduction?

The aim is now consistently stated (pages 2 and 3).

Methods:

Some clarification regarding the sampling, recruitment strategy, and response rates would be helpful. What was the sampling frame (all the schools, or a sub-section?), how many parents were invited to take part (if known), how were they approached, what proportion responded, why was 20 an appropriate number etc.

The remainder of the methods section is very light on detail about data analysis. How was the theoretical framework applied in data analysis, or did it only guide data collection? What were the steps in the thematic analysis? The abstract mentions constant comparison – how was this integrated with the thematic analysis?

Two members of the study team (AB and TS) reviewed the transcripts, coded the interviews independently and then discussed the coding together to identify agreements and disagreements. This led to further refinement of our interpretations (page 5, 130 – 135).

Results:

The authors seem to be presenting the results in three ways: (1) By sub-group (those who don't attend/don't adhere, attend/don't adhere, attend/adhere), (2) according to the six constructs in the theoretical framework (i.e. Tables 2-3), and (3) in terms of themes arising from the data, and broadly related to the theoretical framework (e.g. in the abstract, and on pages 9 and 10). I found this confusing, and at times superficial (particularly 1-2).

I was expecting the results to be presented in terms of the constructs within the theoretical framework.

Could the authors clarify how the reader should 'use' the results section and the two tables? E.g. how the cross-referencing between the three should be done?

Table 3 repeats the quotes in the results section. I can see that it also presents them in terms of the theoretical constructs, so perhaps that is what Table 3 adds. However, this is fairly superficial and could be more convincing. For example – I didn't see how the quotes presented as illustrating self-efficacy actually related to that theoretical construct.

The results section reads like a list, which I think contributes to it appearing superficial. Perhaps it would be helpful to integrate the theoretical contents of table 3 into the text?

Responding to the comments of both reviewers who have commented on the incongruous presentation of the results and how the results do not quite fit into the HBM constructs in Table 3. The results were analysed using thematic analysis however in order to present parental quotes and comply with the word count of the journal the quotes were presented in Table 3. We recognise that this is confusing for the reader and also as the results stemmed from the thematic analysis they do not present an exact fit for the HBM constructs. We have therefore removed Table 3 from the paper and present the quotes in the body of the text.

Line 29, page 7 – I recommend using alternative wording to ‘claimed’, as it implies an inappropriate/unwarranted scepticism about what those parents said.

This has been removed from the revised paper.

Line 27, page 8: “...the benefits of spectacle wear were only reported by parents whose children were adherent.” This seems self-evident, I wasn’t sure what point the authors were trying to make.

This has been removed from the revised paper.

Discussion

The first two pages of the discussion section are actually results (with a few discursive references to wider literature). I think they are the more interpretive, cross-cutting themes that I was wanting above. I recommend that they should be integrated with the results section, bearing in mind the points I have made above. (In other words, the content may need to be re-worked in light of my feedback.)

In response to the reviewers’ comments the discussion section has been revised and the first two pages have been reconfigured into the results section as suggested.

I don’t understand why shared decision making suddenly appears on line 33, page 9 – are the authors over-reaching on what they can claim from their results here?

In the revised manuscript quotes have been presented in relation to the information provided to parents and their understanding of the information. This has specifically been addressed in the section “Implications for clinicians and policy makers” where we suggest that the format and method of sharing information with parents should be considered.

Lines 44-49, page 9. The sentence about parents not appearing to take responsibility is: (a) over-reaching what can be claimed from the data (unless you’re reporting they explicitly said it’s not their responsibility?), and (b) unfortunately worded, in that it sounds like you are blaming parents (which I’m sure is not the case). Could this be re-worded to clarify the analytical/interpretive point being made?

This sentence has been removed from the revised discussion.

I’m struggling with several uses of ‘more likely, less likely’, and the general idea of reporting these results in terms of ‘likelihoods’ (i.e. quantitative connotations, over-stretching what can be interpreted from the data...). Maybe this is inappropriate given the methodology, or maybe it will materialise as entirely appropriate once additional detail is provided on the methods in the study.

We accept the reviewers point and have reworded the text appropriately.

I would like to see a more thorough of strengths and weaknesses at the beginning of the discussion section.

The Strengths and Limitations section has been extended (page 13, line 342 – 360).

Then, more broadly, I'd like to see the discussion section flow more directly from the results, i.e. health beliefs, stigma, and practicalities. For example, the aspects on information provision, communication etc should be discussed in the context of those three main 'themes' in the results.

The structure of the discussion could be clearer, could the authors use the BMJ Paediatrics Open authors guidance?

"Discussion: we recommend, but do not insist, that the discussion section is no longer than five paragraphs and follows this overall structure (you do not need to use these as subheadings): a statement of the principal findings; strengths and weaknesses of the study; strengths and weaknesses in relation to other studies, discussing important differences in results; the meaning of the study: possible explanations and implications for clinicians and policymakers; and unanswered questions and future research."

In light of the reviewers' comments we have reconfigured our results and discussion sections using the journals recommended format. The results are now presented in relation to the three main themes; health beliefs, stigma, and practicalities and Table 3 has been removed from the manuscript with the quotes inserted into the text.

Thank you again, and I look forward to reviewing a re-submission, if required.

Associate Editor

Comments to the Author:

I believe that the authors can respond to the reviewers' critiques and make appropriate changes especially around describing qualitative methods and the health belief model. It would definitely be publishable. I do believe that the title should be changed as both reviewers have suggested

The more comprehensive description of the methodology has been provided and the presentation of the results and discussion revised in light of the reviewers comments. The title has also been changed as requested.

Editor in Chief

Title change to "Semi-structured interviews with parents of children who failed vision screening at the age of 4-5 years: a qualitative study"

What this study adds 3rd point consider changing to something specific which guides the reader to improving adherence

Thank you for your suggested change in the title, we propose a new title and hope this will be acceptable to both the reviewers and the editors.

"A qualitative study investigating the perceptions of parents of children who failed vision screening at the age of 4-5 years."

Point 3 has been revised to be more specific;

"The role of schools as facilitators should be considered when developing interventions to promote spectacle wear in young children."